# Using VR Supermarket for Nutritional Research and Education: A Scoping Review

**DOI:** 10.3390/nu17060999

**Published:** 2025-03-12

**Authors:** Cristiana Amalia Onita, Daniela-Viorelia Matei, Ilie Onu, Daniel-Andrei Iordan, Elena Chelarasu, Nicoleta Tupita, Diana Petrescu-Miron, Mihaela Radeanu, Georgiana Juravle, Calin Corciova, Robert Fuior, Veronica Mocanu

**Affiliations:** 1Center for Obesity BioBehavioral Experimental Research, Department of Morpho-Functional Sciences II (Pathophysiology), “Grigore T. Popa” University of Medicine and Pharmacy, 700115 Iasi, Romania; cristiana-amalia.onita@umfiasi.ro (C.A.O.); dietetician.rnd@gmail.com (D.P.-M.); belosinschi.mihaela-alina@d.umfiasi.ro (M.R.); 2Department of Biomedical Sciences, Faculty of Medical Bioengineering, “Grigore T. Popa” University of Medicine and Pharmacy, 700588 Iasi, Romania; ilie.onu@umfiasi.ro (I.O.); calin.corciova@umfiasi.ro (C.C.); robert.fuior@umfiasi.ro (R.F.); 3Center of Physical Therapy and Rehabilitation, “Dunărea de Jos” University of Galati, 800008 Galati, Romania; daniel.iordan@ugal.ro; 4Department of Individual Sports and Kinetotherapy, Faculty of Physical Education and Sport, “Dunarea de Jos” University of Galati, 800008 Galati, Romania; 5Faculty of Automatic Control and Computer Science, “Gheorghe Asachi” Technical University of Iasi, 700050 Iasi, Romania; elena.chelarasu@yahoo.com; 6Romanian Rugby Federation, 011468 Bucharest, Romania; tupita_nicoleta@yahoo.com; 7Sensorimotor Dynamics Laboratory, Faculty of Psychology and Educational Sciences, Alexandru Ioan Cuza University, 700554 Iasi, Romania; georgiana.juravle@uaic.ro

**Keywords:** virtual reality, augmented reality, nutrition education, food choices, eating behavior

## Abstract

According to “The World Health Organization”, obesity during childhood is directly associated with multiple complications and with an increased risk of the installation of various pathologies. Considering the increase in this pathology among children and teenagers, new instruments of prevention are needed. Virtual reality is an innovative tool that offers several advantages over classical therapies, becoming important in various medical fields, starting from phobia treatment, pain relief, and body image perception to education. This technology has been successfully used to study the influence of virtual cues on behavioral responses and can be useful in nutritional education as well as understanding eating behavior. The objective of this scoping review study is to understand the impact of virtual supermarket exposure on individuals’ food choices and to explore the potential of technology on nutrition education in the general population. It seeks to explore purchasing based on product appearance and placement, food prices, nudging conditions and under-pressure decision making. A manual literature search was conducted using the databases Web of Science, SCOPUS and Google Scholar. Included articles were published between 2012 and 2024 using immersive virtual and augmented supermarket environments as a tool to understand food choices and education. The results showed that using higher immersion can be efficient in understanding food choices, rather than a lower immersive tool. The advantage of immersive virtual reality is highlighted by the sense of presence it offers, compared to other devices, providing a safe, controlled environment for users.

## 1. Introduction

Over the past two decades, obesity has become a global epidemic, with the prevalence steadily rising worldwide. According to “The World Health Organization” (WHO), obesity has increased globally over the past two decades, with over 2.5 million adults classified as overweight and 890 million as obese. Alarmingly, the data reveal a concerning rise in obesity among children aged 5 to 19, affecting approximately 8% (or 160 million) of this population [1].

Obesity can be defined as a medical condition characterized by the excessive accumulation of fat in the body’s adipose tissue, which can lead to various health risks and implications [2]. It is linked to various health issues, including a heightened risk of hypertension, hyperlipidemia, cardiovascular disease, diabetes and some cancers [3]. The condition arises from a combination of factors, including energy imbalances resulting from excessive calorie intake, environmental influences, physiological and genetic predispositions, and social factors [2,4]. Among these, dietary habits play a pivotal role, with unhealthy patterns—such as high consumption of fats, sugars, and fast foods, low fiber intake, and skipping breakfast—being strongly linked to the risk of overweight and obesity [3].

Eating habits and food preferences are often established early in life, shaping lifelong patterns. Studies indicate that children who develop unhealthy eating habits are more likely to carry these behaviors into adulthood, increasing their risk of obesity-related comorbidities [4]. Consequently, understanding eating patterns and promoting healthy eating behavior are major concerns for educating individuals and preventing obesity-related conditions and other comorbidities.

Eating behavior is complex and can be influenced by multiple factors: individual factors (attitude, motivation), social environment (family and community), physical environment (food availability in homes, schools, workplaces, stores), and macro-level factors (food marketing, food systems, and agricultural policies). These influences highlight that dietary choices are not purely individual but shaped by broader social and environmental contexts, emphasizing the need for systemic interventions alongside education. Various strategies, including product placement, pricing adjustments, labeling, and promotions, have been implemented in grocery stores to encourage healthier food choices. It is shown that environmental modifications, such as virtual informational signs and healthy checkout aisles, have a positive influence on eating behavior. Discounts on fruits and vegetables have also been effective in increasing healthier purchases. While these interventions can be effective, their impact can be enhanced when combined with nutrition education [5].

Technology has emerged as a valuable tool in nutrition education and behavioral changes, demonstrating a positive impact on patient engagement and learning. Recent studies used more efficient tools that have been used over the years in education, mobile phones and mobile apps, personal digital assistants, internet-based tools, social media, as well as virtual reality being the main ones [6]. Advancements in technology have recently led to a reduction in the cost of virtual reality, promoting its more widespread use in medical treatments. Virtual reality has been applied in the treatment of obesity and eating disorders, enabling the exploration of alternative realities within the virtual space that would otherwise be inaccessible [7,8,9].

This technology, initially used in clinical applications for treating anxiety disorders and phobias, has expanded to address eating disorders and obesity by creating alternative, controlled environments for behavior modification [10,11,12]. The impact of virtual reality on obesity treatment is particularly notable among younger populations, where immersive environments can effectively influence lifestyle choices and eating habits [4].

Virtual reality systems range from non-immersive and semi-immersive experiences, offering basic 3D visualizations, to fully immersive setups that incorporate head-mounted displays and audio devices for a highly interactive experience [13]. The direct interaction with a virtual scenario that allows for the immersion, or the feeling of being present, was previously described as an interaction that offers the same emotions and reactions associated with the ones from the real world [10].

Cue exposure therapy (CET), which uses controlled exposure to food-related stimuli to reduce cravings and prevent overeating, has demonstrated promise in treating eating disorders and obesity [14]. Traditional CET methods face challenges such as logistical limitations and patient discomfort, but virtual reality offers an alternative by creating realistic, controlled environments that address these barriers. Virtual technology provides distinct advantages over traditional CET methods, such as in vivo or imagery exposure, as demonstrated by Ferrer-Garcia et al. [15] While in vivo exposure—where patients confront real-world scenarios—is often preferred, it comes with significant challenges. These include logistical hurdles like ensuring confidentiality and safety during real-world exposure, managing travel to specific sites, and maintaining control over the environment. Conducting in vivo sessions within a clinician’s office can alleviate some of these challenges but inherently limits exposure to proximal cues while excluding contextual cues [15].

Gorini et al. [16] tested three conditions to evaluate responses to food stimuli. In the Real Food View condition, participants were shown six high-calorie foods for 30 s, with 30 s intervals, during which the foods were covered by red plastic lids. The same protocol was used for The Photograph Slideshow condition used to show images of the same foods, and the Virtual Reality condition, where participants used a head-mounted display (HMD) to interact with a 3D virtual environment. The findings indicate that virtual food is as effective as real food, and more effective than photographs, in eliciting psychological and physiological responses in patients with eating disorders. This highlights the potential of virtual stimuli as a practical alternative to real stimuli for inducing emotional reactions [16].

Virtual immersion enhances the sense of reality, increasing its potential for clinical therapeutic applications [17]. However, head-mounted displays in virtual reality can cause cybersickness, a visually induced motion sickness affecting 22% to 56% of users, with symptoms such as nausea, oculomotor disturbances, and disorientation [18].

One innovative use of virtual technology has involved creating simulated supermarket environments to explore eating behaviors and food choices. This technology can provide valuable information on more effective nutrition education strategies that promote healthier dietary choices. Additionally, virtual supermarkets can serve as tools for teaching nutrition during grocery shopping or analyzing shopping habits, offering a unified and straightforward simulation environment to enhance understanding of how contextual factors influence consumer decisions [19].

Research in this field is recent, and the effectiveness of these technologies in education is not very clear. For the systematic literature review, the following questions have been defined:How can a supermarket environment be utilized to understand food choices, and how can it be used for more effective educational strategies?What is the effectiveness of various technologies that include a supermarket environment?What are the advantages and disadvantages of virtual supermarket exposure?

The first question focuses on understanding the role of virtual reality exposure, images exposure, augmented reality, and mixed reality. The second question presents the study objectives and results from the studies that use a virtual supermarket as an environment to better understand food choices and how technology can impact nutrition education. The last topic emphasizes the advantages and disadvantages of virtual environments.

## 2. Materials and Methods

A manual literature search, illustrated in Figure 1, was conducted using the databases Web of Science, SCOPUS, and Google Scholar. The included articles were published between 2012 and 2024. Various search expressions were used, including (“Virtual supermarket”) AND (“Food choice”), (“Virtual supermarket”) AND (“education”). A total of 35 results were found on Web of Science, 3 results on SCOPUS, and 1297 results using Google Scholar. However, in all databases, many irrelevant results were found within the search. Google Scholar search followed the recommendations from JBI manual, which suggests reviewing till the 5th or 10th for potentially relevant results. In this study [20], Google Scholar results were no longer relevant after the 6th page, so they were not reviewed further. Only articles that used immersive virtual and augmented supermarket environments as a tool to understand food choice and education were included. Literature research included the following stages: title search, abstract, and full-text paper search. Included study designs varied, targeting mostly the adult population, with only a few including children. The studied cases focused on food choice and perception, education, nudging strategies, taxes, and discounts, conducted in multiple countries (The Netherlands, Danmark, United Kingdom, Italy, Switzerland, The United States, Australia and New Zealand).

Articles containing virtual avatars and other virtual environments were discharged. Exclusion criteria were also based on the degree of immersion; studies that used only products exposed to online shopping websites and did not have a three-dimensional-based supermarket were not included. Articles only describing environment applications without implementation and without results presented were also discharged. This search resulted in 1335 articles, from which a total of 1302 were excluded and 33 were included in this study.

Reference management was performed using EndNote 20 (https://endnote.com/?srsltid=AfmBOooOnt-CyjrNVC9vTLK1su5K5WEZA_V33LulPrDiWtELorgNYZo2, accessed 1 March 2024).

Additional papers, such as related review papers and previous paper knowledge, were included to reinforce the studies. This scoping review was conducted following the PRISMA-ScR guidelines, provided as a Appendix A for transparency and reproducibility.

Figure 1 presents a flowchart of the literature search and the selection process. In this literature review, articles were grouped into three subgroups: immersive virtual reality, computer, mobile application or other exposures, and virtual reality vs. other types of exposures.

## 3. Overview of Included Studies

### 3.1. Influence of Various Factors in Healthy Food Choices

External influence has an impact on choices and on the creation of food habits. More dimensions are described as being implicated in the obesity pandemic and influencing dietary intake, as well as physical activity. One of the factors involved is food access [21]. Shopping has been characterized in some studies as being influenced by the availability of the food found on the shelves, price, and also product positioning [22].

Food placements in supermarkets have been a marketing strategy to influence purchases for many years, and studies report that most product placements are used for unhealthy food promotion rather than influencing eating behavior in a positive way [23]. Data show that these factors had an influence on food choices among women. The impact has also been shown on children’s eating choices, with parents reporting difficulty in shopping due to unhealthy products’ positioning. Thus, people who follow an automatic form of decision making have been influenced by the supermarket environment [22].

Price is a well-studied factor that can impact product choices; thus, some policies have tried to restrict the promotion of unhealthy food and beverages [24].

Color perception also plays an important role in food choice; some studies affirm that color can impact behavior and correspond to emotions. In marketing, packaging color has been utilized to influence purchases. Research has shown the different perceptions of color and healthiness: red was minimally associated with health, blue was moderately rated, white low to medium with green, perceived as the most sustainable and healthy.

Supermarkets can have an impact on people’s food choices and can be one of the most common environments in which eating behavior can be improved [25]. To determine whether products are healthy or unhealthy, the quantity of the following can play an important role: saturated fat, salt, sugar, as well as calories per serving [26].

Hence, the supermarket environment can offer a range of information on food perception and can allow us to find the best approach in nutrition education.

### 3.2. Effectiveness of Immersive Virtual Supermarkets in Healthy Food Choices

Verhulst et al. [27] conducted a study using immersive virtual reality to examine how the shape of fruits and vegetables affects purchasing behavior. Participants’ perceptions of quality were influenced by the level of abnormality in the produce. The study included four groups: a control group of 35 participants who viewed standard supermarket produce, 36 participants who saw “slightly” misshapen items, 35 participants who were shown moderately misshapen produce, and 36 participants who viewed heavily misshapen items. In the virtual environment, participants walked through a store, located the fruit and vegetable shelf, and chose items as desired, although they were not required to make a purchase. Overall, the results showed little difference in purchasing behavior, regardless of the degree of deformity in the produce [27].

In recent years, virtual reality scenarios have become valuable tools in various research that emphasize the role in understanding and promoting eating behavior based on food choices. For instance, Lombart et al. [28] used a virtual grocery store with the purpose of investigating if food imperfection can affect fruit and vegetable purchases as well as exploring VR effectiveness on consumer behavior. The results showed a preference for visually perfect products, and factors such as awareness of food waste, environment, and price influenced the purchase process. It concluded that, while aesthetics plays an important part in food purchases, educational information through various campaigns may change the perspective on food exterior aspects [28].

Health education through virtual supermarkets was studied by Smit et al. using biscuits as the main products. In this, the participants were children, between 6 and 13 years old, with an average of 9 years old. Participants completed a training session where they were instructed to explore the virtual supermarket, select a product, and place it into a shopping basket. Once the training finished, the next stage of the exposure started. Subjects were verbally instructed to search for the fruit and biscuits shelves, where they could choose one preferred option. When choosing the product, a pop-up image and text appeared, which were linked to health or environmental appeal. In the fruit biscuit category, two options presented no risk, two of them with medium risk, and two involved high risk based on the consumption consequences shown in the images and text. Before the product selection, participants were asked to carefully review each pop-up. After the choice was made, products were placed in the shopping basket. All participants recalled the images shown for each product chosen. The subject’s age is important when it comes to understanding this information. In conclusion, this approach increased awareness about the impact of behavior [4].

Food choice, using a virtual store, was also studied by Hall et al. involving parents of children aged 1 to 5. Participants were assigned to view one of the fruit drinks displaying no artificial sweeteners, 100% Vitamin C and 100% all natural. The control group was exposed to drinks with no claim. Participants were able to visualize all sides of the 3D drink images, including the nutritional information. When rotating the product, subjects could also view the label. In the virtual environment, subjects were guided to the beverage coolers, where they were asked to choose from two options. One task was designed to choose between a healthier 100% juice and a less healthy drink (two grape-flavored drinks, a fruit drink labeled according to the assigned condition and 100% grape juice with no label claims). In the second task, a selection was made involving different types of beverages (one apple-flavored fruit drink and plain water with no claim). This was considered an “across-category” decision, as it involved products from different beverage types. The results of the study showed that claims viewed by the subjects led to an incorrect belief that the fruit drinks contained no added sugar and were 100% juice compared to the control group [29].

Schnack et al. [30] examined shopper personality and how it impacted behavior using the Big Five personality traits (agreeableness, conscientiousness, extraversion, neuroticism, and openness to experience). The research included a shopping experiment conducted in a 3D virtual store, where participants imagined they were invited to a social gathering and had to purchase snack items, such as crackers and soft drinks. Additional items could have also been purchased. The store included 19 product categories, including chocolate bars, biscuits, chewing gum, sweets, muesli bars, crackers, potato chips, ice cream, canned beans, canned tomatoes, canned corn, bread, milk, flavored milk, soft drinks, tea, juice, and magazines. The findings showed no significant impact of shopper personality on the investigated purchase behaviors, indicating a need for further research [30].

Another approach of virtual reality was viewed in Blitstein et al. [31], where, this time, food package nutritional labels were used in order to examine food decisions among adults. The adult selection was made based on the following inclusion criteria: having at least 1 child between the ages of 4 and 12 and having a low income. The subjects were divided into two groups, one that had only 10 min to shop and one with no time restraint. There were three label designs that participants were exposed to, a summary label with visual stars that indicated how healthy the product is, products that presented stars on the front of the package that provided information about calories per serving, called the hybrid label, and a nutrient-specific label that showed calories per serving and percent daily values for sugar, total fat, saturated fat, and sodium.

The shopping task included six products that were identified as foods purchased in low-income households, such as bread, breakfast cereal, granola bars, potato chips, crackers, and frozen pizza.

All packages with food labels increased healthy food choices compared to packages without information. Simpler food labels, such as the hybrid one, led to a healthier nutrient profile compared with the nutrient-specific package label. Time had an impact as well; adults exposed to simple food information and with time pressure chose less healthy products compared to the ones introduced to nutrient-specific front-of-package nutrition labels [31].

Food choices have been analyzed in various studies. Some research included eye tracking, replicating a real-life situation, and making it possible to understand the complexity of the food selection process. This system captures visual stimuli from the environment, such as shape and color.

Melendrez-Ruiz et al. [32], in research on 120 participants, with ages between 20 and 65 years, analyzed food choices. In the virtual environment, French supermarket food items were integrated, without displaying prices or expiration dates.

The products were placed in different areas of the grocery store, including a total of 48 different food items from four food groups: proteins, carbohydrates, vegetables/legumes, and lipids.

To analyze the participants’ eating behavior, different scenarios were created, including daily choices, choices based on how healthy a product is considered, choices made based on the environment, and based on the foods that participants prefer to consume.

Results for the first and last scenarios showed a noticeable preference for animal-based products, compared to legumes.

Participants’ choices based on the environment resulted in a high preference for vegetables and animal-based products, with legumes and cereals being the least chosen food groups. Regarding the health scenario, participants predominantly chose vegetables over all other food groups. The final results of this study showed the close relationship between food choices and the scenarios presented to participants [32].

Schnack et al. [33] conducted a study using a virtual store that featured products across 15 different food categories, such as potato chips, chocolate, ice cream, and canned goods. Participants received a written shopping list and were placed in a hypothetical scenario where they imagined a friend or family member needed items for guests from a nearby convenience store. Each product category included three to eight brands with various flavors, brands, prices, and shelf positions. The results showed that shoppers exhibited realistic behaviors, including a tendency to purchase a reasonable proportion of private-label brands. They also selected more items from higher shelf positions, made impulsive purchases, and spent less time examining familiar brands compared to unfamiliar ones. Additionally, women were found to spend more time in store, handle products for longer periods, and make higher overall expenditures than men. These findings indicate that participants demonstrate authentic shopping behaviors in immersive virtual store simulations, suggesting that such environments can serve as a cost-effective alternative for studying consumer behavior [33].

Cekatauskaite et al. [34] focused on food positioning in the checkout aisle on health purchases. Data were collected after a virtual supermarket exposure as well as a real supermarket, by applying questionnaires. The results showed that subjects did not realize the product positioning at the checkout aisle. On the other hand, multiple participants confirmed that product visibility on aisles would positively influence grocery choices, selecting more healthy choices [34].

Virtual reality was also used as a tool to understand eating behavior under pressure. Nudging, defined as influencing choices without forbidding any options, has been used to analyze food choices under time pressure in a virtual supermarket. In Goedegebure et al.’s [35] research, participants were asked to purchase four products from different food groups, provided in a shopping list. When immersed in the virtual environment, it could be seen that, in total, 20 product categories were used in the virtual supermarket. The package’s illustration did not include additional information about the product. The nudging condition in this research was focused on making prominent healthier choices from each food group, resulting in positively influencing subjects.

Healthier light products can face a taste disadvantage compared to regular products due to reduced fat, sugar, or salt. The virtual store environment was built using 360° images. For the immersion, participants used a virtual reality headset and a smartphone. When interacting with the virtual store, a pop-up appeared when the subject focused on a product, providing details as well as the product. Three conditions were included: no cue, regular popular product, and light popular product. Subjects included in the study experienced one category where no popularity information was provided, one where a regular product was marked as popular, and one where a light product was marked as popular. The virtual experience started at the entrance, where some choices were made at the vegetable and peanut aisles. The next task was choosing from the following options: cheese, chocolate milk, and hot dogs. The results showed that popularity cues increased the selection of light products but had no effect on regular product choices [35].

Blom et al. [36] used an immersive supermarket to study food choices under pressure. Participants were immersed in the virtual supermarket and were asked to buy four products listed on a grocery shopping list. The required items included a dessert, a bottle of soda, pasta, and cheese. Within each product category, participants could choose one specific item and could also navigate through the supermarket by walking or using a teleportation feature. Once the product selection was made and all four products were in the baskets, the task was completed, and participants could proceed to answer the prepared questionnaire on the computer. Participants in the under-pressure condition were instructed to make these choices in 3 min to identify the nudging effect. The results indicated similar experiences with impulsive and reflective decision making being the same in both scenarios [36].

### 3.3. Computer, Mobile Application or Other Exposures in Food Choice Intervention

Intentions for food shopping are evaluated differently when shopping is planned. While individuals with a strong intention to shop healthily prioritize health when preparing shopping lists, once in the supermarket, decisions are often influenced by additional factors. Understanding taste’s impact on food choices while shopping was studied by Mergelsberg et al. [37] A total of 113 female undergraduate students were recruited for the study and were asked to simulate food shopping behavior. This task assessed speed in selecting or rejecting food products based on their healthiness and tastiness using a computerized supermarket scenario. Audio cues of either “healthy” or “tasty” were received, and a joystick was used to select or reject the various options. The results showed that the selection of tasty foods was faster, regardless of healthiness, and only a few participants selected healthy products quickly. The findings suggest that tastiness continues to influence food choices, especially for individuals with a large gap between their healthy shopping intentions and eating behavior [37].

Finding ways to understand the eating behavior of children and adults with obesity was researched by Jayachandran et al. [38]. A game was developed that introduced a more realistic supermarket experience to 30 subjects between the ages of 17 and 29. This game had various steps, such as asking participants to create a shopping list, selecting the general information (e.g., name, age, and gender), and entering a supermarket. In the shopping environment, the subject could explore it using a keyboard and add the products that were previously written on the list to a basket. The game was designed to offer adaptive levels depending on their choices, helping to educate the participants in regard to what products are considered healthy or unhealthy. Using this computerized game, players received education regarding food products selected in everyday life [38].

In order to help support healthier grocery choices, Ahn et al. [39] created a mobile augmented reality application that promoted personalized recommendations, highlighting products that are healthy and ones to avoid for specific health-related issues, such as allergies, as well as information about low-sodium or low-fat diets and adequate caloric intake. This application was based on color tagging, green to indicate nutritionally adequate food and red to show products to avoid. The color tags appeared while shopping, with the role of increasing the desire of 15 participants to find healthier products and to gain more speed in choosing food [39].

Mizdrak A et al.’s [40] study aimed to understand food purchasing behavior across different income groups, including a total of 98 participants; 47% of participants completed the study. Low-income participants were significantly less likely to complete the study.

During the shopping tasks, participants were instructed to imagine they have no food or drink in the house, and they are going to the supermarket to buy all the food and drinks needed. The second shopping task was framed as occurring a week after the first.

The objective of the study was to evaluate variations in purchasing behavior and the response to the virtual computerized supermarket, particularly across different income groups. The findings suggest that the virtual environment is a promising tool for examining food purchasing behaviors.

The grocery list approach was also used by Eykelenboom et al. as a tool to examine food purchases, focusing on price relevance. In this study, participants, exposed to a virtual store, were assigned to three groups: the control group was exposed to products with regular prices, one was exposed to a sugar-sweetened beverage tax, and the last group to nutrient profiling tax conditions based on Nutri-Score. A scale of five-point color codes is indicated by the Nutri-Score. It starts from the healthiest products (dark green or the letter ‘A’) to the least healthy (red or the letters ‘D’ and ‘E’). In this case, the results showed an improvement in the participants’ basket, when it comes to the healthiness level of the products and total calories, for the participants exposed to the nutrient profiling tax conditions based on Nutri-Score [41].

Nudging and price strategies were used by Van der Mole et al. [42] in a computerized supermarket, where 400 Dutch participants were introduced to five study conditions (control, nudging, pricing, salient pricing, and salient pricing with nudging).

Budgets were based on subjects’ weekly grocery spending. They could only check out if their total spending was between 50% and 125% of their budget. Every week, participants were exposed to different conditions: the control condition was similar to a Dutch supermarket, the nudging condition used orange arrows and frames to highlight healthier options, and the pricing condition adjusted the prices of various products (a 25% increase in unhealthy products and a 25% decrease in healthy products). The salient pricing condition added information about price changes, while the final condition combined nudging and salient pricing. The results were inconclusive. Limited evidence of nudging and pricing strategies was found in food purchasing behaviors. The results suggest the possible implementation of these strategies in supermarkets for health promotion [42].

Stuber et al. [43] used a three-dimensional web-based virtual supermarket to investigate the effects of nudging and pricing strategies on food purchases in The Netherlands, exploring the individual and combined effects of nudging, taxes with a 25% price increase, and a 25% price decrease, across various food categories, including fruits and vegetables, grains, dairy, protein products, fats, beverages, snacks, and other foods. Compared to the control condition, the combined approach of subsidies on healthy products and taxes on unhealthy products, within the nudging and price salience condition, proved to be the most effective. Healthy purchases increased by 9% for fruits and vegetables, 16% for grains, and 58% for dairy. Purchases of protein products and beverages remained unchanged. Unhealthy products and grain purchases decreased by 39% while dairy decreased by 30%. There were no significant changes in snack or beverage purchases. Substitution toward healthier options was observed within the grain and dairy categories.

Hoenink et al. [44] designed a supermarket to simulate a real-life shopping experience by imitating a typical Dutch supermarket, including 1200 products categorized into 12 food groups. The proportion of healthy products in the virtual supermarket was comparable to the proportion found in real-life supermarkets, with 19% of items considered healthy, compared to 16% of products in Dutch supermarkets. In the study, participants were exposed to five conditions: control, nudging, pricing, price salience, and price salience with nudging They were exposed to an increase of 25% in unhealthy products, a 25% discount on healthy products, or a 25% price increase and discount. Thus, 455 participants of low and high socioeconomic positions, with 49.2% having high education and 32.8% reporting high income, conducted their weekly shopping in the virtual supermarket for five consecutive weeks. The study’s primary outcome was the percentage of healthy products purchased per week, understanding the impact of pricing and nudging on healthy and unhealthy purchases. The second outcome included the total grams of healthy and unhealthy products purchased and understanding how this influenced food behavior.

Nudging and non-salient pricing strategies did not significantly increase healthy food purchases, a combination of salient price increases and discounts significantly increased the percentage of healthy food purchases, and combining salient pricing with nudging strategies increased healthy product purchases.

Waterlander et al. conducted multiple studies [45,46,47,48] using a three-dimensional virtual supermarket to examine the effects of price interventions on food purchasing behavior. Participants, recruited through various methods, completed weekly grocery shopping tasks at home using virtual budgets based on their income.

One study [45] focused on understanding the impact of sugar and sweetened beverage taxes on beverages and snack purchases, where 120 participants were included in two experimental conditions. One had products with a 19% tax, and the second condition, or the control one, had standard pricing on all sugar-containing drinks, including soft drinks, fruit juice, flavored milk, and energy drinks. The shopping experience allowed participants to navigate aisles and select items with a computer mouse. At the end of the shopping task, participants proceed to the store checkout, filling out a final questionnaire. The findings showed a lower purchase in the price increase condition compared to those in the control group, suggesting that increasing the tax rate reduced sweetened beverage purchases without affecting the consumption of other products.

Price intervention on other products was studied, introducing taxes on fruits and vegetables, on sweetened beverages, on saturated fat, salt, and sugar. The study involved adults who were able to speak English and could connect to a computer or laptop. Participants were asked to purchase groceries for the upcoming week, and 1412 food items were displayed on shelves, with a selection derived from the top-selling products in New Zealand. Products had labeled prices that appeared when the cursor was over the item. To complete the shopping task, a virtual checkout was created where participants paid for their groceries [46].

Waterlander et al. conducted another study using a virtual supermarket, where most participants completed a shopping task. Each participant had their task at home and was asked to plan a weekly grocery shop. Information about their income was provided, which was used to allocate a shopping budget. Participants navigated through the aisles using the cursor keys, selecting products with a single mouse click and adding the items to a shopping cart. Individual prices and the total expenditure were listed. Participants could access product nutritional information by clicking on an information icon and could remove items if needed. Upon completing their shopping, participants proceeded to the checkout, where they were directed to complete a closing questionnaire.

The study had two conditions: the control condition with regular pricing and the experimental condition where a 25% discount was applied on fruits and vegetables. The discounted items included fresh, frozen, and canned fruits and vegetables. Using Excel’s Random Number Generator, participants were randomly assigned to either the control or experimental group. The results showed that a 25% discount on fruits and vegetables significantly increased their purchases by 984 g per household per week, indicating a 25% rise and a price elasticity of demand of 1.0. Discounts did not increase expenditure on less healthy foods and did not influence the total calories [47].

Another study that aimed to understand the impact of discounts and food labels on purchasing products used the three-dimensional supermarket. The study included 18-year-old or older participants, fluent in Dutch, with a lower socioeconomic status, and with lower education or unemployed. Participants completed a typical weekly household shopping task in the virtual supermarket with an allocated budget. The primary outcome measures included the number and proportion of healthy and unhealthy items purchased, the total grams of fruits and vegetables, and total calories. The proportion of more nutritious products within specific food categories was also calculated. Price perception, habits, quality, participants’ awareness of prices, and label were measured using a 7-point Likert scale. The results showed an increase in healthy food purchases with a 50% discount, compared to 10% and 25% discounts. It was also found that higher discounts resulted in greater total energy intake, and no significant effects were found for the different labels [48].

Pini et al. [49] investigated the impact of augmented reality on consumers’ food choices in two experiment groups, making a comparison between traditional shopping and augmented shopping. Participants’ food selections, time spent choosing, and information behavior were analyzed. The results showed that augmented reality highlighted nutritional information, helping consumers make healthier choices by downplaying front-of-package claims. The experiment, conducted in a controlled lab environment, involved products like cereals and granola bars displayed without price labels. Two categories of products were chosen, such as cold breakfast cereals and granola bars. Each product front package claimed nutritional information, such as “light”, “palm oil-free”, “organic”, “high in fiber”, and “high in vitamin content”. Participants using augmented reality relied more on nutritional information than packaging appearance. The study concluded that this technology can enhance consumer awareness of a product’s nutritional value, promoting healthier decisions [49].

### 3.4. Virtual Reality vs. Other Supermarket Exposures

Interactions with various virtual immersions can show the impact these technologies have on users’ choices. De Vries et al. [50] studied food choices using a touchscreen for images and food choices using a mouse. Products for the research were selected from a Dutch supermarket, selecting a low-sugar dairy drink and a low-fat one. During the test, participants completed the study protocol for both environments.

The 360° format scoring was significantly higher on the touchscreen compared to the 2D images. In conclusion, the research provided positive evidence of touchscreen interaction, having an effective response from consumers. Thus, a more immersive and interactive environment can influence food choices.

More recent studies focused on finding new tools to enhance nutrition education, remarking on the high prevalence of overweight in Western countries. Thus, some research used mixed-reality environments in order to understand which tool has better results in promoting healthy food choices.

All technologies can be useful in education, and many studies have compared the effectiveness of shopping in a real store, using virtual reality or 2D exposure [51].

Waterlander et al. [52] compared a virtual supermarket and real exposure in a 3-week study period. Both real and virtual scenarios were conducted in the same weeks, the virtual task being completed at home by each subject. Participants were asked to plan their groceries for the following week, as they planned in real life. In order to understand participants’ food patterns, they completed a questionnaire regarding shopping habits. The computerized virtual supermarket offered the possibility to navigate around the store with the computer mouse. Measures for this research studied the expenditure for the 18 food categories in both scenarios, the number of products chosen, as well as the feeling of presence in the virtual store. The results confirm the similarity in food purchases in the virtual supermarket and the real one when it comes to selection and budget. Participants also experienced a feeling of presence using the computerized store; only those with a higher capacity to focus perceived a stronger feeling in the moment [52].

Siegrist et al. [53] studied the different reactions people have in a virtual exposure and a real one. The focus of this study was on the non-habitual decisions of the participants, compared to other research. Participants were assigned to complete two tasks, either in the real world or a virtual environment. The real-life scenario was located in a laboratory-controlled environment, exposing 33 cereal options on a shelf. For both environments, two tasks were required. The first one focused on the selection of cereals more suitable for a children’s camp with ages between 10 and 12. In the second task, participants were instructed to choose a cereal for a friend who follows a low-sugar diet, a selection that first required checking the nutritional information. The results indicated that a virtual environment that offers interaction with products can be used as a tool to better understand food choices [53].

Some studies compared the feeling of presence in an immersive environment and a scenario used on a desktop system. In this direction, Schnack et al. [54] studied the level of telepresence in both scenarios. In the first experiment, participants navigated the virtual computerized store using a mouse and keyboard. For the immersive virtual reality technology, a head-mounted display, body tracking sensors, and hand-held motion-tracked controllers were used, allowing participants to explore the virtual environment. Subjects completed their task, purchasing groceries in each virtual simulated store. The results revealed greater immersion for subjects exposed to the immersive virtual reality store and more natural interactions compared to the desktop scenario [54].

Di Fang et al. [53] exposed participants to either a non-hypothetical or hypothetical experiment using various presentations of the products, such as text, pictures, and a virtual grocery store. Actual yogurt labels with nutritional information were created and applied to real packages of light and original strawberry yogurt. The results indicate that the hypothetical bias in virtual reality was not significantly different from text or picture. Virtual reality significantly reduced some participants’ hypothetical bias in choice experiments. The conclusion highlighted that incorporating a level of realism through immersion in a virtual environment could enhance the realism of the food choice [55].

Another study that used a real and immersive supermarket was conducted by Pizzi et al. [56], measuring and comparing levels of hedonism, utilitarianism, store satisfaction, as well as perceived assortment size. Participants were exposed to both physical and virtual stores. In virtual and real conditions, participants were exposed to a bakery shelf and were given the freedom to choose the products. Price was also standardized for these scenarios. In the virtual setting, participants used a virtual headset and controllers, which provided direct interaction with the virtual products. The interaction was also seen in the results, which showed similar behavior across both store types [56].

A comparison between low immersion and highly immersive virtual environments was also studied by Woodall et al. [51]. Participants were immersed in a supermarket scenario that replicated a medium-sized supermarket. Nutrition information could be accessed, and purchases could be made using voice commands. The environment included realistic supermarket sounds, such as background announcements and cash register noises. For the lower virtual immersion, a PC monitor was used. The study results show that participants reported a greater sense of presence in the immersive scenario compared to the PC scene. Heart rates and electrodermal activity were also recorded, showing significant changes during the virtual exposure compared to the real scene [51].

Van Herpen et al. [57] explored consumer behavior across three product categories, such as fruits and vegetables, milk, and biscuits. The number, variety, and type of products selected were examined, as well as the money spent and responses to price promotions and shelf displays in pictures, real, and virtual stores. The findings show that behavior in the virtual store is more similar to the real store compared to the picture. Participants purchased more products and spent more money, responding strongly to price promotions in both virtual and pictorial environments compared to the physical store [57].

Liu et al. [58] studied a real store using audiovisual presentations for product sampling. For the virtual grocery store, a video was shown with a grocery store’s peanut butter aisle shelf, capturing, through audio, the ambient sounds of the grocery store, including conversations and the noise of carts being pushed through the aisle. Identical peanut butter samples were shown, with labels either hidden, revealed, or revealed during the study, highlighting the all-natural status of one peanut butter sample. A photo of the peanut butter labels was used, showing labeled and unlabeled jars. The results showed that the all-natural label enhanced the perceptions of the product quality compared to the regular sample. The virtual shopping environment boosted the consumer perceptions of quality and nutrition products [58].

In conclusion, virtual reality has a positive effect compared to 2D or other exposures. One of the benefits of virtual exposure is the feeling of presence that participants have, feeling similar to real-life experiences. Thus, behavior can be positively influenced by using a virtual environment [57,59].

## 4. Negative Effects of Virtual Reality

Virtual immersion heightens the sense of reality, enhancing the interest in using virtual reality clinically as a therapeutic intervention [17]. Virtual reality with head-mounted displays can lead to side effects known as cybersickness, exhibiting symptoms similar to motion sickness, or described as a type of visually induced motion sickness. This condition can occur in between 22% and 56% of cases and is marked by symptoms like nausea, oculomotor disturbances, and disorientation when exposed to virtual environments [18].

Cybersickness is often assessed using various questionnaires, showing the difference between non-immersive and immersive exposures. The results affirm that non-immersive content shown on a television screen resulted in the lowest scores on a simulation sickness questionnaire, while the highest scores were observed with immersive content viewed through a head-mounted display [60].

Typically, visual, vestibular, and somatosensory cues work together to establish self-motion perception, but while using virtual reality tools, conflicts among these cues can result in cybersickness, possibly influenced by the sensory information for postural control [18].

Another study suggests that some virtual environments present inconsistent focal distances across objects, requiring continuous accommodation without changes in eye vergence, which can lead to visual discomfort. Additionally, it is often associated with side effects like dizziness, headaches, and nausea due to mismatches between the visual information and sensory input from the body rather than disturbances in the vestibular or proprioceptive systems [17].

Many studies have tried to observe these symptoms, and it has been observed that cybersickness symptoms are experienced both during and after virtual exposure. The difference between motion sickness and cybersickness can be explained by the level of movement. Thus, cybersickness occurs while the user is usually stationary and experiences a strong sensation of movement due to shifting visual imagery. The length of exposure and exposure repetition are significantly correlated with increased symptoms of sickness. For instance, some studies have shown a positive interaction when repeating the exposure to the visual content, reducing sickness motion [61].

Many factors are suggested as influencing cybersickness, such as age, gender as well as the application used. Thus, aside from individual differences, system design, such as virtual, technical limitations, like image distortions and inconsistent visual cues, can contribute to these symptoms. Research indicates that younger individuals tend to be more resistant to simulation sickness; after age 40, the vestibular perceptual threshold declines, increasing susceptibility to cybersickness. Findings also suggest a correlation between gender, with some women experiencing higher levels of simulation sickness than men. Other studies found that females were equally prone to motion sickness.

In conclusion, cybersickness can result from various software, hardware, and environmental factors, impacting user comfort and the practical application of virtual reality. Thus, it is important to focus on cybersickness and its causes, identifying the contributing factors and offering guidelines for minimizing cybersickness symptoms [59,60].

## 5. Results and Discussion

Table 1, Table 2 and Table 3 provide a descriptive summary of virtual and augmented interventions that introduce three-dimensional supermarket scenarios and tools that ultimately could be used in food choice education. Researchers have studied the role of supermarkets from various perspectives, including the influence of labels and product appearance, as well as choosing under pressure and rewarding shopping. Adult consumers mostly formed the studied population, with only a few including adolescents and children. The types of studies were experimental designs, quasi-experiments, qualitative and quantitative methods, randomized trials, cross-over design, exploratory studies, mixed randomized experiments, as well as interview studies. Table 4 highlights the main focus of the studies (Food Choice and Perception, Education, Nudging, Taxes and Discounts).

This study aimed to review the literature on a virtual reality intervention in education and food choices. It describes different levels of intervention: immersive virtual reality, computer, mobile application, or other exposures, and virtual reality vs. other types of exposures. We identified 33 primary studies addressing food behavior using various strategies, such as food shape perception, food choices based on item position, education, and the impact of taxes on food purchases.

This study also highlights that individual eating behavior is shaped by a complex interaction of various factors across different contexts. Many interventions prioritize individual responsibility, often overlooking the socio-cultural and economic environments that strongly influence consumer decisions. Food choice is influenced by availability, pricing, and marketing strategies, all of which play a significant role in shaping purchasing behavior [62].

Most studies that aim to understand food choices in supermarkets include understanding the consumer’s behavior for different food groups, the advantages of shopping stores for educational purposes, the variety of products and the possibility of comparing labels, and comparisons based on participants’ income. Many studies included in this review utilized immersive virtual reality, augmented reality, or image-based approaches to examine the impact of supermarket scenarios on food choices, particularly in the context of health promotion. Our findings indicate that both immersive and computerized supermarkets can improve food choices; however, when comparing these two types of exposure, immersive environments show a greater sense of presence and improve the perception of quality and nutrition products compared to computerized 3D supermarkets. These findings have significant practical implications for public health, nutritionists, and dietitians. A better understanding in food purchasing can help design more effective educational tools, ultimately improving decision making related to eating behavior and health. Using real supermarkets has presented barriers for research, due to many factors, such as lack of trust from stores, the flexibility of price changes, and changing shelf layouts. Other disadvantages are the high prices involved, the longer time needed for data collection, limited control as well as the impossibility of replicating the study. Creating a supermarket replica in a more controlled environment could be a solution but, in some cases, could be a challenge due to financial costs. Thus, virtual grocery stores may be a good approach to expose people to a more controlled environment, created using a real-life store. The advantages of virtual reality can also be seen in some studies that compared these tools. Woodall, S. studied which environment creates a better feeling of presence, a PC supermarket and restaurant scenario or an immersive virtual environment. The feeling of presence was also studied, using questionnaires as well as heart rate and electrodermal activity. The supermarket environment offered the possibility of reading the nutrition labeling, as well as purchasing the products. Background noises were also included, such as people’s conversations, announcements, and the sound of cash registers. In the restaurant, participants could access the menu and purchase food, without having nutritional information. The tasks were completed in all the different scenarios, and virtual environments indicated a greeted feeling of presence compared with the other scenario [51].

## 6. Conclusions and Future Directions

The popularity of immersive virtual reality supermarkets has grown over the years due to their benefits in understanding food choices and in promoting nutrition education. One significant advantage of immersive virtual reality is the higher feeling of presence offered compared to devices like phones, tablets, or PCs, which provide only limited immersion. Virtual reality enables full user immersion, an advantage that can positively influence the effectiveness of the method, offering a safe controlled scenario for users. One notable limitation is the risk of cybersickness, which can interact with the performance of the user, limiting data research. This condition encompasses a range of symptoms, including nausea, disorientation, and other discomfort that some users may experience while interacting with virtual environments. Immersive virtual supermarkets can be used in further research to promote healthy food choices, including a larger number of participants and age groups. This review has several strengths, providing a literature synthesis on how external factors can influence eating behavior. It offers a global perspective on how technology can be a useful tool in education, highlighting various practical methods for further research. Data filters applied during the search process may have excluded relevant studies with valuable insights on virtual supermarket interventions. Further research in this field is still needed considering broader search criteria.

## Figures and Tables

**Figure 1 nutrients-17-00999-f001:**
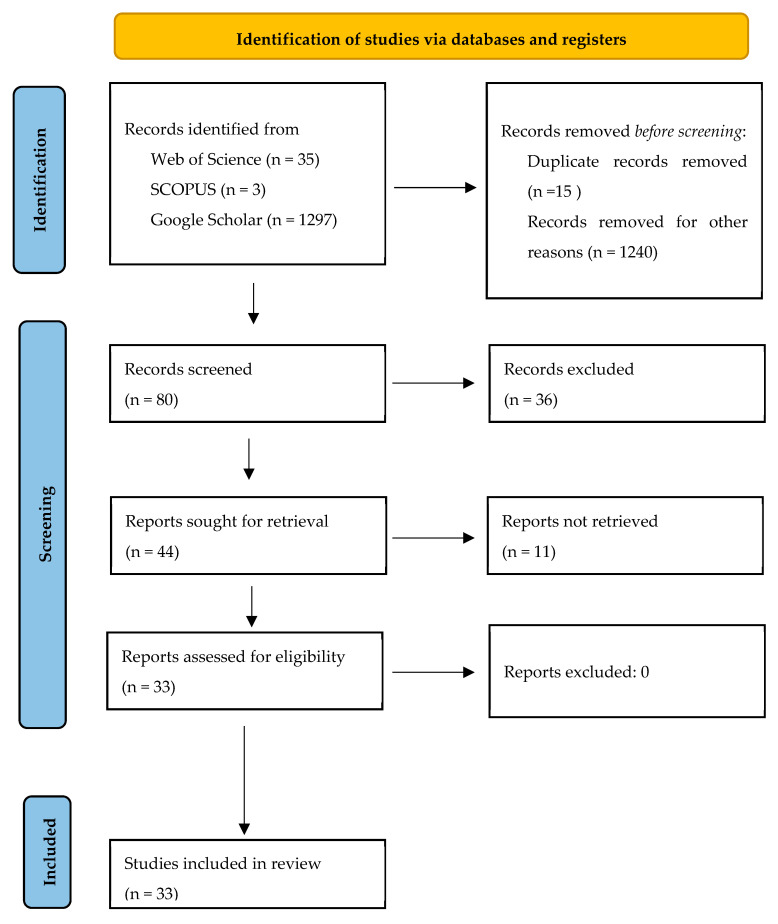
Flowchart of literature search and selection process.

**Table 1 nutrients-17-00999-t001:** Summary of the basic characteristics of studies that used immersive virtual supermarkets. These studies were published from 2012 to 2024.

Immersive Virtual Reality						
No.	Author	Country, Year	Studied Population (Sample Size)	Types of Studies	Objective	Results
1	Blitstein J.L. et al. [31]	United States, 2020.	Adult consumers (parents) (*n* = 1452)	Experimental design	To examine the impact front-of-package nutrition labels (FOPLs) have on decision-making abilities among low-income parents in a virtual supermarket.	Simple front-of-package nutrition labels help parents in selecting healthier products compared with nutrient-specific labels.
2	Blom, S. S. A. H. et al. [36]	The Netherlands, 2021.	Adult consumers (*n* = 99)	Quasi—experiment	Understanding if the effect of nudging on healthy food choice is enhanced under time pressure by introducing participants to a virtual supermarket.	No differences was determined by price and nudge in decision-making experiences, showing that individuals have similar experiences with impulsive and reflective decision-making.
3	Cekatauskaite V. et al. [34]	Danmark, 2015	Adult consumers (*n* = not specified)	Qualitative and quantitative methods	Understanding consumer perception of food positioning and the impact on choices based on the checkout aisle.	Study suggested that participants were not aware of the healthier products placed in the checkout aisle; several participants claimed that they would be likely to purchase more healthy groceries if they were more visible to them.
4	Goedegebure R. P. G. et al. [35]	The Netherlands, 2020	Adult consumers (*n* = 300)	Quasi-experiment	Understanding if food brands influence purchase and stimulate consumers to choose healthier products.	Consumers are more likely to choose light products when combined with popular cues. However, this had no effect on regular food choices.
5	Hall M. G. et al. [29]	United States, 2022	Adult consumers (Parents) (*n* = 2219)	Randomized trial	Understanding whether nutrition information on fruit drinks impacts purchasing among parents.	Claims viewed by the subjects led to an incorrect belief that the fruit drinks contained no added sugar and were 100% juice.
6	Lombart C. et al. [28]	France, 2019.	Adult consumers (*n* = 142)	Experiment	The effects of fruits and vegetables aesthetics on food purchasing behavior.	A preference for visually perfect products was observed, factors such as awareness of food waste, environment and price, had influenced the purchase process.
7	Melendrez-Ruiz J. et al. [32]	France, 2022.	Adult consumers (*n* = 120)	Quasi-experiment	Understanding food choices and motivation using eye-tracking in a virtual supermarket.	It was shown the relationship between food choices and the scenarios presented to participants.
8	Schnack A.et al. [30]	New Zealand, 2021	Adult consumers (*n* = 113)	Quasi-experiment	Understanding shopper personality and how it impacted shopper behavior using an immersive virtual reality convenience store.	Shopper personality did not show any impact on the investigated purchase behaviors. Further studies are needed.
9	Schnack, A. et al. [33]	New Zealand, 2019	Adolescents and adult consumers (*n* = 153)	An exploratory study	Understanding purchase based on brand product popularity.	There was a higher selection of products from higher shelf positions spending less time examining familiar brands compared to unfamiliar ones.
10	Smit E.S. et al. [4]	The Netherlands, 2021	Children consumers (*n* = 22)	Interview Study	Health education in children and studying environmental impact of food consumption using a virtual store.	All participants recalled the images shown for each product chosen, the subject’s age has an importance when it comes to understanding the information. The results, increased awareness about the impact of behavior.
11	Verhulst A. et al. [27]	France, 2017	Adult consumers (*n* = 142)	Experiment	Understanding purchase based on fruits and vegetable shape and feeling of presence in a virtual environment.	A similarity in food purchases whatever their deformity was observed. Perceptions of fruit and vegetable quality depended on the level of abnormality.

**Table 2 nutrients-17-00999-t002:** Summary of the basic characteristics of studies that use computer, mobile application or other types of exposures.

Computer, Mobile Application or Other Exposures						
No.	Author	Country, Year	Studied Population (Sample Size)	Types of Studies	Objective	Results
12	Ahn J. et al. [39]	United States, 2015.	Adult consumers (*n* = 119)	Quasi-experiment	Augmented reality mobile application that makes healthy recommendations in real-time and highlights products for various types of health concerns: allergies to milk or nut products, low-sodium or low-fat diets, and general caloric intake.	Reduces the amount of time it takes for shoppers to find desired healthy food products and avoid unhealthy ones.
13	Eykelenboom M. et al. [41]	The Netherlands, 2022	Adult consumers (*n* = 394).	Randomized controlled trial	The effects of sugar-sweetened beverages and nutrient profiling tax, based on Nutri-Score, on consumer food purchases using a virtual supermarket.	Basket healthiness was higher, with a lower energy content, 3301 kcal/week for households, for participants exposed to the nutrient profiling tax based on Nutri-Score (color-coded for most healthy foods (dark green, associated with the letter ‘A’) to least healthy (red, associated with the letter ‘D’ or ‘E’)
14	Hoenink, J. C. et al. [44]	The Netherlands, 2020	Adult consumers (*n* = 455)	Randomized experiment	Understanding how nudging and price can increase healthy food purchase	Combining salient pricing and nudging strategy increased the percentage of healthy product purchases.
15	Jayachandran K. et al. [38]	United States, 2017.	Adult consumers (*n* = 30)	Experiment	Obtain more information regarding shopping patterns.	It was showen the potential of educating the subjects and convincing them to purchase healthier food products.
16	Mergelsberg E.L.P. et al. [37]	Australia, 2018	Adult consumers (*n* = 68)	Quasi-experiment	Studying the perception of products, based on health perception or tastiness using supermarket shelves photos during the shopping task.	Taste was considered a more important motivator, a result associated with the relative speed of food selection and rejection. Selection based on healthiness was not as rapid.
17	Mizdrak A. et al. [40]	United Kingdom,2017	Adult consumers (*n* = 98)	Quasi-experiment	Assessing the feasibility of a virtual supermarket to measure food purchasing behavior in different income groups.	Low-income participants were significantly less likely to complete the study, only 47% of participants completed the study.
18	Pini V. et al. [49]	Italy, 2023	Adult consumers (*n* = 50)	Laboratory experiment	Understanding the effects of delivering nutritional information using augmented technology on food choices.	AR technology facilitated the choice of healthier food items. Additionally, participants in the experimental group reported that they based their decisions on nutritional information rather than on the appearance of the package.
19	Stuber, J. M. et al. [43]	The Netherlands, 2021	Adult consumers (*n* = 318)	Mixed randomized experiment	Examining which food groups were most impacted by nudging and pricing strategies.	Healthy purchases increased for fruits, vegetables grains and dairy (9%, 16%, and 58%)
20	Van der Molen A. E. H. et al. [42]	The Netherlands, 2021	Adult consumers (*n* = 400)	Exploratory study	Examination of purchase motivation, after nudging, price	Data was inconclusive. Limited evidence of nudging and pricing strategies was found in food purchasing behaviors. The results suggest the possible implementation of these strategies in supermarkets for health promotion.
21	Waterlander W. E. et al. [48]	The Netherlands, 2013	Adult consumers (*n* = 109)	Experiment	The influence of price discounts and food labels on promoting healthier food purchases.	Price discounts had a stronger influence on purchasing behavior than food labels
22	Waterlander W. E. et al. [47]	The Netherlands, 2012	Adult consumers (*n* = 197)	Randomized trial	Understanding the impact of fruit and vegetable discounts on purchases	A 25% discount on fruits and vegetables increased their purchases
23	Waterlander W. E. et al. [46]	New Zeeland, 2019	Adult consumers (*n* = 1193)	Randomized experiment	Understanding price intervention on fruits and vegetables, sweetened beverages, saturated fat, as well as salt, and a sugar tax	The sweetened beverage tax and fruit and vegetable tax intervention showed non-significant impacts on food purchases, with changes of 0.18% and 0.41%. Saturated fat and salt taxes led to significant substitution effects, increasing fruit and vegetable purchases by 4.0% and 4.3%, and raising sugar intake as a percentage of total energy by 5.0% and 3.2%.
24	Waterlander W. E. et al. [45]	New Zeeland, 2014	Adult consumers (*n* = 120)	Randomized controlled trial	Understanding the impact of sugar and sweetened beverages taxes on beverages and snack purchases.	Increasing the tax rate reduced sweetened beverage purchases without affecting the consumption of other products.

**Table 3 nutrients-17-00999-t003:** Summary of the basic characteristics of studies that use virtual reality vs. other supermarkets of exposures.

Virtual Reality vs. Other Supermarkets of Exposures						
No.	Author	Country, Year	Studied Population (Sample Size)	Types of Studies	Objective	Results
25	de Vries R. et al. [50]	The Netherlands, 2018	Adult consumers (*n* = 50)	Cross-over design	Investigating the degree of perception using—direct touchscreen and indirect touch using a mouse.	It showed the role of object interaction in shaping behavior using a touchscreen device.
26	Fang D. et al. [55]	United States, 2020.	Adult consumers (*n* = 256)	Experiment	Investigate whether virtual reality can reduce hypothetical bias in choice experiments, exposing them to text, pictures, and a virtual grocery store.	For participants with low simulator discomfort, results suggest that virtual reality can significantly reduce hypothetical bias in choice experiments.
27	Liu R. et al. [58]	United States, 2017	Adult consumers (*n* = 120)	Experiment	Understanding the evaluation of identical products, with only one labeled as all-natural. In a separate condition, the all-natural label was highlighted to assess the influence on consumer evaluations.	It was indicated the enhanced consumers’ perception of product quality and nutritional content of the all-natural label compared to the regular sample. There was no significant impact on purchasing.
28	Pizzi G. et al. [56]	Italy, 2019	Adult consumers (*n* = 50)	Quasi-experiment	The objectives are testing consumer perceptions (satisfaction, perceived assortment size), shopping orientations (utilitarian and hedonic), and behaviors (choices, time spent in front of the shelf) in virtual and real stores.	Virtual reality was highly associated with realism. It was reported a lack of physical interaction between the participant and the products.
29	Schnack A., et al. [54]	New Zealand, 2018.	Adult consumers (*n* = 111)	Experiment	Understanding the level of telepresence in an immersive supermarket compared to a desktop store.	A greater immersion was ibserved for subjects exposed to the immersive virtual reality store and more natural interactions compared to the desktop scenario.
30	Siegrist M. et al. [53]	Switzerland, 2019	Adult consumers (*n* = 68)	Experiment	Understanding choices made in front of a real supermarket shelf and a virtual one.	Virtual condition simulated a real environment in which participants can interact can be a useful tool for conducting experiments related to food choices.
31	van Herpen et al. [57]	The Netherlands, 2016.	Adult consumers (*n* = 90)	Experiment	Understand the difference in food choices in various environments.	It was shown a similar behavioral response in the virtual reality scenario with the choices made in the real store, compared to the picture condition.
32	Waterlander W. E. et al. [52]	New Zealand, 2015	Adult consumers (*n* = 123)	Validation study	Comparison between a virtual supermarket and a real-life scenario.	Purchase patterns in the virtual environment were similar to the real experience. The difference in purchase was seen for the following: fresh fruit and vegetables, dairy, and snack foods.
33	Woodall S. et al. [51]	United States, 2024	Adult consumers (*n* = 21)	Experiment	Understanding food choices using nutritional labels and understanding the feeling of presence.	The virtual environments indicated a greeted feeling of presence compared with the other scenario.

**Table 4 nutrients-17-00999-t004:** Summary of the studied responses of the 33 included studies.

No.	Publication		Used Cases	
		Food Choice and Perception	Education	Nudging	Taxes and Discounts
1	Blitstein J.L. et al. [31]	√	√		
2	Blom, S. S. A. H. et al. [36]	√	.	√	
3	Cekatauskaite V. et al. [34]	√		.	
4	Goedegebure R. P. G. et al. [35]	√	√		
5	Hall M. G. et al. [29]	√			
6	Lombart C. et al. [28]	√			
7	Melendrez-Ruiz J. et al. [32]	√			
8	Schnack A. et al. [30]	√			
9	Schnack, A. et al. [33]	√			
10	Smit E.S. et al. [4]	√	√		
11	Verhulst A. et al. [27]	√			
12	Ahn J. et al. [39]		√		
13	Eykelenboom M. et al. [41]				√
14	Hoenink, J. C. et al. [44]			√	√
15	Jayachandran K. et al. [38]	√	√		
16	Mergelsberg E.L.P. et al. [37]	√			
17	Mizdrak A. et al. [40]	√			
18	Pini V. et al. [49]	√	√		
19	Stuber, J. M.et al. [43]			√	√
20	Van der Molen A. E. H. et al. [42]			√	√
21	Waterlander W. E. et al. [48]		√		√
22	Waterlander W. E. et al. [47]				√
23	Waterlander W. E. et al. [46]				√
24	Waterlander W. E. et al. [45]				√
25	de Vries R. et al. [50]	√			
26	Fang D. et al. [55]	√			
27	Liu R. et al. [58]	√			
28	Pizzi G. et al. [56]	√			
29	Schnack A., et al. [54]	√			
30	Siegrist M. et al. [53]	√			
31	van Herpen et al. [57]	√			
32	Waterlander W. E. et al. [52]	√			
33	Woodall S. et al. [51]	√

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
