# Peer review of "Using VR Supermarket for Nutritional Research and Education: A Scoping Review"

_nutrients, 2025, doi:10.3390/nu17060999_

Round 1

Reviewer 1 Report (New Reviewer)

Comments and Suggestions for Authors

The review addresses an interesting topic since the use of new technologies must also be transferred to nutritional education. My comments are directed at three specific points:
a) Both in the introduction and in the discussion, reference should be made to the need to combine educational actions for food selection with environmental policies and actions. There is a growing tendency to focus on individual responsibility when environmental responsibility is as important or more important. It should be reflected in the article in a more accentuated and highlighted way that these actions with new technologies require environments that favor these actions and not just delegate responsibility to the consumer.
b) Table 1: in the "results" column, eliminate the word "results" since they are assumed in the title of the column.
c) In conclusions, the main conclusions of the review should appear.

Author Response

Comment 1. Both in the introduction and in the discussion, reference should be made to the need to combine educational actions for food selection with environmental policies and actions. There is a growing tendency to focus on individual responsibility when environmental responsibility is as important or more important. It should be reflected in the article in a more accentuated and highlighted way that these actions with new technologies require environments that favor these actions and not just delegate responsibility to the consumer.

Response:

We have revised both the Introduction and Discussion sections. Introduction (Page 2, Paragraph 69-81), Results and Discussions (Page 95, Paragraph 764-769):

Comment2. Table 1: in the "results" column, eliminate the word "results" since they are assumed in the title of the column.

Response

We have revised the Table eliminating the word "results" (Page15-25)

Comment 3. In conclusions, the main conclusions of the review should appear.

Response

The conclusion section was modified in the Conclusions and future directions section (Page 96, Paragraph 820-826)

Reviewer 2 Report (New Reviewer)

Comments and Suggestions for Authors

-In order to enhance the clarity and reporting quality of the paper, it would be advisable for authors to use the PRISMA checklist and upload it as a supplementary file. Although scoping reviews do not have and dedicated checklist, many organizations suggest the use of PRISMA in such studies. 

- When using PRISMA, the authors would be asked to show the search strategy EXACTLY as conducted, so that other researchers could copy and paste the search strategy and verify the reproducibility of the search. This would enhance transparency and the quality of the present study. Also, the dates of the last search must be included in the methods section. 

The authors claim to have used only Google Scholar in the abstract, but the main text says they also used Scopus and Web of Science. Please correct this.

The use of date filters is not advisable since important studies must be removed from the search. I suggest authors include this limitation or provide an important justification for using it. 

Lines 137-138: To enhance objectiveness, I suggest citing a reference (such as https://jbi-global-wiki.refined.site/space/MANUAL/355861885/5.2.4.3+Searching+for+grey+literature) to support the Google Scholar search up to the sixth page.

-  Figure 1: Please use the PRISMA flowchart in order to enhance the clarity and reproducibility of the paper. 

The authors should split Table 1 into different tables, maybe according to the type of VR (“computer mobile application and other exposures” should be one table, and “immersive VR” should be another). Such tables should be on a page in landscape format, which would greatly facilitate reading. The columns “studied population” and “type of study” should become separate. Authors should present the sample size of each study in a column (maybe along with “study population”). 

The authors should consider merging the results and discussion sections since the results section is already highly discursive, and the discussion section is extremely succinct in this paper.

minor

  • there is a break between lines 51 and 52 that was mistakenly placed there;
  • lines 134 and 135 and figure 1: these are results and not methods of the paper. please readjust.
  • Lines 148: “Through previous paper knowledge”. I am sorry I did not understand what this means, could the authors please clarify?
  • line 666: would not be better “outcomes” than “responses”?
  • line 687: authors must spell out the first name of the authors from such references. It is inadequate to use “(19)(20)...”

Author Response

Comment 1.-In order to enhance the clarity and reporting quality of the paper, it would be advisable for authors to use the PRISMA checklist and upload it as a supplementary file. Although scoping reviews do not have and dedicated checklist, many organizations suggest the use of PRISMA in such studies.

- When using PRISMA, the authors would be asked to show the search strategy EXACTLY as conducted, so that other researchers could copy and paste the search strategy and verify the reproducibility of the search. This would enhance transparency and the quality of the present study. Also, the dates of the last search must be included in the methods section.

Response

We uploaded as a supplementary file the PRISMA-ScR = Preferred Reporting Items for Systematic reviews and Meta-Analyses extension for Scoping Reviews.

Comment 2.-The authors claim to have used only Google Scholar in the abstract, but the main text says they also used Scopus and Web of Science. Please correct this.

Response

We corrected the abstract (Page 1, Paragraph 36)

Comment 3.-The use of date filters is not advisable since important studies must be removed from the search.

I suggest authors include this limitation or provide an important justification for using it.

Response

The use of ate filters was included as a limitation in the Conclusions and future directions section (Page 97,Paragraph 823)

Comment 4.-

Lines 137-138: To enhance objectiveness, I suggest citing a reference (such as https://jbi-global-wiki.refined.site/space/MANUAL/355861885/5.2.4.3+Searching+for +grey+literature) to support the Google Scholar search up to the sixth page.

Response

We have added in the Materials and Methods section (Page 4, Paragraph 158)

Comment 5.-- Figure 1: Please use the PRISMA flowchart in order to enhance the clarity and reproducibility of the paper.

Response

We have used the PRISMA flowchart (Page 4)

Comment 6.-The authors should split Table 1 into different tables, maybe according to the type of VR ("computer mobile application and other exposures" should be one table, and "immersive VR" should be another). Such tables should be on a page in landscape format, which would greatly facilitate reading. The columns "studied population" and "type of study" should become separate.

Response

We have separated the main table into 3 different tables (Table 1, Table 2, Table 3) (Page 15)

Comment 7.-Authors should present the sample size of each study in a column (maybe along with "study population").

Response

Sample size was added along with "study population"

Comment 8.-The authors should consider merging the results and discussion sections since the results section is already highly discursive, and the discussion section is extremely succinct in this paper.

Response

The Resuls and discussion sections were merged Results and Discussions section (Page 95)

Reviewer 3 Report (New Reviewer)

Comments and Suggestions for Authors

The authors have taken up a very interesting topic but did not avoid significant errors and omissions in their work. Below are a few of them:

1) the goal of the study is not clearly formulated. I do not understand the provisions relating to nutritional education (lines 32, 112, 121, 128 - "...it promotes nutritional education"), 

2) the methodology does not contain key information that must be included in the scoping review
- data collection date (this information is in the abstract, but it should also be in the methodology because it is key)
- the authors' procedure recorded in line 136 is not clear
- the selection of studies - the procedure presented in Figure 1 is unclear - it is unknown whether the authors considered studies/articles or references. How many of them were finally included? The description of the results shows that 33 and Figure 1 show that 33+28 (?)
- a structured description of the studies taking into account elements that are important from the point of view of the research question, which will enable comparisons of what was the essence of the review (such a description should be included in the methodology so that the reader knows what the presentation of the results will be)

3) discussion of the results - in my opinion, there was a lack of presentation of the study results in practical terms - what practical implications does the review have, and where and by whom can this knowledge be applied

ln this first line of discussion (line 670), the authors wrote: "...This study aimed to review the most recent literature." - it is not true because of the authors in the line 33 wrote." ... published between 2014 and 2024 using". 

There was no indication of this scoping review's strengths and limiting factors - which, in my opinion, is extremely important because it proves the ability to be self-critical and proves scientific maturity.

Author Response

Comment 1. the goal of the study is not clearly formulated. I do not understand the provisions relating to nutritional education (lines 32, 112, 121, 128 - "..it promotes nutritional education",

Response

We modifies the lines 32, 131-132, 141,148)

Comment 2. the methodology does not contain key information that must be included in the scoping review

- data collection date (this information is in the abstract, but it should also be in the methodology because it is key)

- the authors' procedure recorded in line 136 is not clear

. - the selection of studies - the procedure presented in Figure 1 is unclear - it is unknown whether the authors considered studies/articles or references. How many of them were finally included?

The description of the results shows that 33 and Figure 1 show that 33+28 (?)

- a structured description of the studies taking into account elements that are important from the point of view of the research question, which will enable comparisons of what was the essence of the review (such a description should be included in the methodology so that the reader knows what the presentation of the results will be)

Response

We added data collection in the Materials and Methods (Page 3, Paragraph 150)

Line 136 was deleted.

We have used the PRISMA flowchart (Page 4)

We have included a more structured description in the methodology section (Page 4, Paragraph 162)

Comment 3.  discussion of the results - in my opinion, there was a lack of presentation of the study results in practical terms - what practical implications does the review have, and where and by whom can this knowledge be applied

In this first line of discussion (line 670), the authors wrote: "...This study aimed to review the most recent literature." - it is not true because of the authors in the line 33 wrote." ... published between 2014 and 2024 using".

There was no indication of this scoping review's strengths and limiting factors - which, in my opinion, is extremely important because it proves the ability to be self-critical and proves scientific maturity.

Response

We modified and presented the practical implications in the Results and Discussions aria (Page96, Paragraph 778)

We deleted the phrase – the most recent (Page 95, Paragraph 762)

We added the strength and limitations in the  Conclusions and future directions section (Page 97, Paragraph 818)

Round 2

Reviewer 3 Report (New Reviewer)

Comments and Suggestions for Authors

I thank the authors for introducing the corrections according to my suggestions. Currently, the article is much better, including the methodology. In my opinion, it requires minor corrections, the description is as follows:
1) since the review included 33 articles, the record in lines 174-176 about the additional 28 works is not clear (?) I do not understand what this is about.
2) the results chapter no. 3 was removed but points 3.1 - 3.4 remained
3) the next chapter no. 4 is "results and discussion" (?) I do not understand

Author Response

We appreciate the reviewer’s perspective. We have revised the manuscript sections.

Comment 1. Since the review included 33 articles, the record in lines 174-176 about the additional 28 works is not clear (?) I do not what this is about.

Response:

We have revised lines 174-176 and deleted it.

Comment2. The results chapter no. 3 was removed but points 3.1 – 3.4 remained

Response

We have modified adding the section 3. Overview of Included Studies (Page5) and added section 4. Negative effects of virtual reality.

Comment 3. The next chapter no. 4 is results and discussion (?) I do not understand

Response

Based on the previous suggestions, the two sections were merged. 5. Results and Discussions (Page 25)

This manuscript is a resubmission of an earlier submission. The following is a list of the peer review reports and author responses from that submission.

Round 1

Reviewer 1 Report

Comments and Suggestions for Authors

This article conducts a comprehensive review of research on virtual reality supermarkets in the fields of food choice and nutrition education. It emphasizes the potential applications of virtual reality technology in obesity prevention and promoting healthy eating, with particular attention to children and adolescents, providing new ideas and methods for solving the obesity problem.

However, there are the following relatively serious problems:

Limitations of research methods:

Single database: Only using Google Scholar for literature search may miss relevant research in other databases, resulting in insufficient comprehensiveness of research results. It is recommended to use cross-search and comparison results of multiple databases.

Unclear establishment of literature database: In lines 138 to 140 on page 3, the authors' indicate that they delete 97 papers from 126 papers and retains 26 papers. However, the total of 97 + 26 is not equal to 126. Where did the remaining three papers go? In addition, the authors mentioned finding 14 additional papers. Thus, the total number of papers analyzed is 40. But these data do not correspond to the total of 50 in Figure 1. This inevitably makes people doubt the rigor of the authors in handling the original literature.

Subjectivity of exclusion criteria: The subjectivity of some exclusion criteria (such as studies based on product exposure on online shopping websites) may affect the objectivity and repeatability of the research. For example, online shopping websites may also introduce AR or VR technology on product pages, and these all require more in-depth exploration.

Insufficient depth of result analysis:

Lack of integrated analysis: The discussions on various research results are mostly descriptive and do not deeply analyze the internal connections and potential mechanisms between different research results, making it difficult to form a systematic theoretical framework.

Insufficient exploration of influencing factors: Factors influencing food choice (such as consumer personality, cultural background, etc.) have not been fully discussed, limiting the explanatory power and universality of research results.

Reviewer 2 Report

Comments and Suggestions for Authors

Although this research deals with a potentially relevant topic (application of immersive technologies on consumer science), it presents several issues with the research goal and the applied methods. In particular, the methods present several flaws, which severely hinder the scientific quality of the paper:

- Only using one database (Google Scholar), which is not ideal for conducting a literature review. Instead, authors should have used other databases, such as Scopus and Web Of Science (ideally both).

- The applied queries are not clear. Did the authors use three different expressions (Virtual supermarket AND Obesity; Virtual supermarket tool treatment of obesity; Virtual reality grocery list AND food choices)?  Or did they use more than those three expressions? Why did authors apply several different expressions (and thus, conducted several parallel searches) instead of applying a single expression ("Virtual supermarket" OR "Virtual reality") AND (obesity OR "food choices")?

- The lack of a scientifically accurate search method is reflected in the source of paper inclusion: of the 40 papers included in this review, 14 (35%) were references not included in the database search. This indicates that either the applied query was unsuitable for the review or there were issues with the article inclusion/exclusion (which is not likely since 126 references were analyzed).

- Why was the search limited to articles published between 2014 and 2024?

- How was a degree of immersion used as an exclusion criteria?

- How were the additional papers included? Was it through the authors' previous knowledge? Analysis of related review papers? Analysis of the references list of the included studies?

- Figure 1 is also incongruent with the numbers reported in the text (a total of 40 references with 14 additional papers).

Furthermore, it is not clear what were the inclusion criteria:

- Exactly, which type of immersive technologies were considered? Virtual reality, mixed reality, videowall/ledwall, representation on mobile devices?

- Which type of consumer behaviors were considered for inclusion? 

Considering these flaws, the paper should be rejected in its current form. The methodology should be substantially improved.

Reviewer 3 Report

Comments and Suggestions for Authors

This article provides valuable insights into the application of virtual reality technology in nutrition education but there is still room for improvement in terms of scope and depth of research. Specifically, the following three suggestions are proposed for adoption and modification by the author: 1. It is recommended that the author consider including studies from different databases to ensure the comprehensiveness of the research. 2. It is suggested to provide more data to support the conclusions regarding the negative effects of virtual reality. 3. It is recommended to offer specific recommendations for future research in the conclusion section to promote further development in this field.